# Think Less, Act Early: Reinforced Latent Reasoning with Early Exit in Vision-Language-Action Models

**Dianqiao Lei** [1]   **Lianlei Shan** [1]

## Abstract

Existing Vision-Language-Action (VLA) models predominantly rely on explicit Chain-of-Thought (CoT) reasoning to bridge perception and action. While effective, this paradigm suffers from high computational costs and error propagation in multi-step tasks. In this paper, we propose *Adaptive Variable Alignment VLA (AVA-VLA)*, a novel *Latent Reasoning VLA framework* that models reasoning as a sequence of unobservable latent variables, bypassing the need for explicit text generation. However, latent trajectories are inherently susceptible to noise interference and misalignment with downstream objectives. To address this, we introduce a *Reinforcement Learning-based Denoising mechanism* that treats latent state generation as a sequential decision process, optimizing reasoning trajectories via task-level rewards. Furthermore, we incorporate an *Early-Exit Strategy* that adaptively terminates reasoning based on state confidence, enabling a dynamic trade-off between depth and efficiency. Extensive experiments on embodied decision benchmarks demonstrate that *AVA-VLA* significantly reduces inference latency while achieving superior stability and success rates compared to full-reasoning baselines.

## 1. Introduction

Vision-Language-Action (VLA) models, by unifying visual perception, language understanding, and action decision-making, have demonstrated significant potential in embodied intelligence and complex sequential tasks (Driess et al., 2023). To bridge the semantic gap between high-dimensional observations and low-level executable actions,

[1]Tsinghua University, Beijing, China. Correspondence to: Lianlei Shan <shanlianlei18@mails.ucas.edu.cn>.

*Proceedings of the $43^{rd}$ International Conference on Machine Learning*, Seoul, South Korea. PMLR 306, 2026. Copyright 2026 by the author(s).

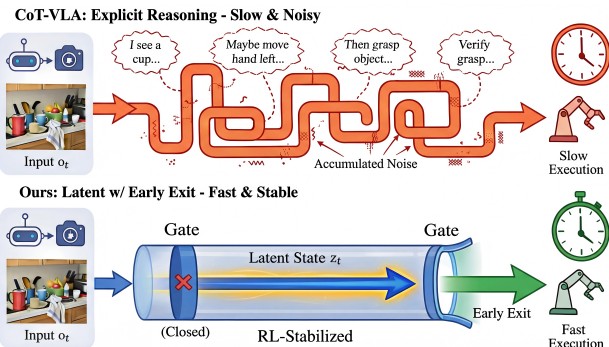

*Figure 1.* **Explicit Chain-of-Thought vs. Reinforced Latent Reasoning.** *Top*: Explicit CoT paradigms suffer from high computational latency and error propagation (accumulated noise) due to discrete token generation. *Bottom*: Our approach models reasoning as a continuous Latent State evolution stabilized via RL. The Early-Exit mechanism (green arrow) adaptively terminates the latent dynamics to achieve a superior trade-off between inference speed and decision stability.

these models typically require complex reasoning capabilities. However, achieving efficient and stable reasoning while ensuring decision quality remains a core challenge in current VLA research.

Recent approaches have predominantly adopted *explicit reasoning* paradigms, such as generating Chain-of-Thought (CoT) or step-by-step plans, to enhance interpretability and performance (Zhao et al., 2025; Qu et al., 2025; Wei et al., 2022). While effective in certain scenarios, the limitations of this "unfolded" reasoning are evident, as illustrated in **Figure 1 (Top)**. First, generating complete reasoning texts incurs significant computational overhead, making it difficult to meet the low-latency demands of real-time robotics. Second, multi-step explicit reasoning is prone to *accumulated noise*—errors in early reasoning steps propagate and amplify, causing intermediate states to deviate from task goals. Furthermore, these methods often rely on manually designed text supervision, limiting their generalization to unspoken, intuitive physical skills.

To address these limitations, we propose a perspective distinct from the explicit paradigm: treating reasoning as *continuous latent dynamics* optimized directly for the final task

goal. Specifically, we introduce *Adaptive Variable Alignment VLA (AVA-VLA)*, a novel framework that models reasoning not as interpretable text, but as a sequence of unobservable latent variables. As shown in **Figure 1 (Bottom)**, this latent evolution guides cross-modal integration and action generation without the cost of token-by-token decoding, offering greater flexibility in modeling complex thought processes.

However, moving from explicit to implicit reasoning introduces a new challenge: without direct text supervision, the latent trajectory is susceptible to representation drift and noise. To strictly align the latent space with downstream objectives, we introduce a *Reinforcement Learning (RL)-based Denoising mechanism*. By modeling latent generation as a sequential decision process, we optimize the reasoning policy via task-level reward signals (e.g., success rate, trajectory consistency). This ensures that the latent states remain expressive yet stable, effectively "denoising" the thought process.

Furthermore, in practical embodied scenarios, reasoning demands vary dynamically—simple tasks require fast reactions, while complex ones demand deep deliberation. Inspired by this, *AVA-VLA* incorporates an *Early-Exit Reasoning Strategy*. A gating mechanism monitors the confidence of the latent state, adaptively terminating the reasoning process to generate actions immediately when the state is sufficient. This enables a dynamic trade-off between reasoning depth and computational cost.

In summary, this work systematically redesigns the VLA reasoning mechanism through implicit modeling, RL optimization, and adaptive computation. Our main contributions are:

- We propose *AVA-VLA*, a framework that models reasoning as a sequence of latent variables, bypassing the computational bottleneck and stability issues of explicit CoT.

- We introduce an *RL-based Reasoning Denoising* method, which treats reasoning as a decision process and optimizes latent trajectories via task-level rewards to ensure alignment and stability.

- We design an *Early-Exit Strategy* that enables the model to adaptively switch between "thinking fast" and "thinking slow," optimizing the trade-off between inference efficiency and performance.

- We validate *AVA-VLA* on multiple multimodal reasoning and embodied decision benchmarks, demonstrating that it significantly reduces inference latency while achieving performance comparable to or better than full-reasoning baselines.

## 2. Related Work

**Vision-Language-Action Models and Architectures.** VLA models unify perception and control by grounding language into robotic actions. Building on PaLM-E (Driess et al., 2023) and RT-2 (Zitkovich et al., 2023), recent works have focused on specialized architectures to enhance manipulation capabilities. For instance, $\pi_0$ (Black et al., 2024) introduces flow matching to unify discrete and continuous control, while Octo (Team et al., 2024) and OpenVLA (Kim et al., 2024) employ diffusion transformers and action chunking for generalist policies. To improve multi-task transfer, UnifiedVLA (Wang et al., 2025b) and UniVLA (Bu et al., 2025) leverage large-scale video generative pretraining. Others like VLA-Adapter (Wang et al., 2025a) and TraceVLA (Zheng et al., 2024) focus on parameter-efficient fine-tuning or utilizing trajectory history. While these methods significantly improve control precision, they typically rely on the inherent capacity of the base model without explicitly modeling the *internal reasoning process* required for complex decision-making.

**Explicit Reasoning and Multimodal Chain-of-Thought.** To handle long-horizon tasks, integrating Explicit Reasoning (Chain-of-Thought) has become a mainstream direction. CoT-VLA (Zhao et al., 2025) and SpatialVLA (Qu et al., 2025) fine-tune models to generate explicit reasoning steps or spatial descriptions before acting. NORA (Hung et al., 2025) retrieves relevant experiences to guide reasoning, while WorldVLA (Cen et al., 2025) and FLOWER (Reuss et al., 2025) explicitly predict future states or world dynamics to assist planning. Although explicit CoT enhances interpretability and robustness (Wei et al., 2022), it introduces significant inference latency due to long-context decoding. Furthermore, Turpin et al. (Turpin et al., 2023) highlighted the "unfaithfulness" risk, where generated text diverges from actual policy decisions.

**Implicit Reasoning and Latent Dynamics.** Implicit reasoning aims to bypass the "language bottleneck" by performing deliberation in latent space. In the VLA domain, predictive models (Tian et al., 2024) and interactive tuning methods (Tan et al., 2025) utilize latent representations or test-time interventions to correct policies without explicit text generation. In the broader LLM context, Quiet-STaR (Zelikman et al., 2024) and Coconut (Hao et al., 2024) treat reasoning as continuous latent state evolution. Our approach aligns with this trend but differs fundamentally by treating the latent reasoning process as a *reinforcement learning problem*. Unlike World Models (Hafner et al., 2019) that focus on dynamics prediction, we optimize the latent trajectory specifically for decision confidence and task success.

**Reinforcement Learning for Cognitive Optimization.** Reinforcement Learning (RL) in VLA is evolving from simple action fine-tuning (Schulman et al., 2017) to optimizing

cognitive processes. Following the success of DeepSeek-R1 (Guo et al., 2025) in incentivizing reasoning via RL, recent works like VLA-R1 (Ye et al., 2025) apply verifiable rewards to align reasoning with manipulation goals. We extend this by applying RL not just to final actions or output text, but to the *internal latent reasoning steps*, effectively "denoising" the thought process to ensure stability and alignment with robotic control.

**Efficient Inference and Adaptive Computation.** Efficiency is critical for real-time robotics. PD-VLA (Song et al., 2025) accelerates inference via parallel decoding to increase throughput. In the LLM field, Early-Exit mechanisms (Chen et al., 2023) have been used to dynamically reduce computation. However, most existing VLA efficiency methods (like PD-VLA) focus on the decoding head. Our work introduces an Early-Exit mechanism applied directly to the *latent reasoning stream*, allowing the model to adaptively terminate internal deliberation based on uncertainty estimates, offering a trade-off between reasoning depth and reaction speed that is distinct from decoding acceleration.

## 3. Method

This section systematically elaborates on the proposed *Adaptive Variable Alignment VLA (AVA-VLA)* framework from a formal modeling perspective, as illustrated in **Figure 2**. We first define the problem setting and notation, then cast the latent reasoning process as a *Reasoning-centric Partially Observable Markov Decision Process (POMDP)*. Based on this formulation, we introduce the RL-based reasoning denoising mechanism and the adaptive early-exit strategy.

### 3.1. Problem Definition and Overall Modeling

We consider a standard multimodal sequential decision-making problem. At each time step $t$, the agent receives a multimodal observation tuple from the environment:

$$o_t = \{v_t, l_t, h_{t-1}\}, \tag{1}$$

where $v_t$ denotes the visual input, $l_t$ represents language instructions or context, and $h_{t-1}$ captures the history of interaction states (e.g., encodings of past observations or actions). Based on this observation, the model is required to generate an action $a_t \in \mathcal{A}$ to maximize the long-term task reward.

Distinct from traditional methods that map directly from observations to actions, we explicitly introduce an **unobservable latent reasoning state** $z_t \in \mathcal{Z}$ within the model. This state characterizes the intermediate reasoning representation of multimodal information during the decision process. Crucially, this latent variable does not require an interpretable semantic structure; instead, it serves as an implicit state dependent on the decision-making process,

learned end-to-end via task objectives.

### 3.2. POMDP Formulation of Latent Reasoning

In Vision-Language-Action tasks, the reasoning process bridging multimodal perception and action decision-making is inherently internal and not directly observable. Although visual inputs and language instructions are explicit observations, the mechanism by which the model integrates historical information, maintains task goals, and gradually forms a decision basis relies on the evolution of implicit intermediate states. This process exhibits clear temporal dependencies and uncertainty.

Treating latent reasoning merely as a forward computation or static intermediate representation implies a failure to capture its dynamic evolution and impact on long-term decision goals. Therefore, we propose to view the generation of latent reasoning as a sequential decision problem controlled by a policy, naturally modeling it as a **Partially Observable Markov Decision Process (POMDP)**. This perspective allows us to characterize the evolution mechanism of reasoning states under a unified probabilistic framework, providing a theoretical basis for introducing RL optimization and adaptive reasoning depth control.

Specifically, we formalize the evolution process of latent reasoning as the following POMDP tuple $\mathcal{M} = (\mathcal{Z}, \mathcal{O}, \mathcal{U}, P, R, \gamma)$:

- **Latent Reasoning State Space ($\mathcal{Z}$):** $z_t \in \mathcal{Z}$ describes the internal reasoning state formed by the model at time step $t$. It is used to encode the integration of multimodal information, task execution progress, and abstract representations related to decision-making.

- **Observation Space ($\mathcal{O}$):** This corresponds to the multimodal observations $o_t = \{v_t, l_t, h_{t-1}\}$ received by the agent from the environment, containing encodings of visual inputs, language instructions, and interaction history.

- **Latent Reasoning Update Action Space ($\mathcal{U}$):** In this POMDP, $\mathcal{U}$ represents the space of latent reasoning update actions. Unlike environmental interaction actions, a *reasoning update action* $u_t \in \mathcal{U}$ acts solely within the model, controlling how the latent reasoning state is updated.

- **Transition Dynamics ($P$):** Given the current latent state $z_t$, multimodal observation $o_t$, and reasoning update action $u_t$, the latent reasoning state evolves according to the conditional transition distribution:

$$z_{t+1} \sim P(z_{t+1} \mid z_t, u_t, o_t). \tag{2}$$

This transition process characterizes how the model modifies and extends existing reasoning states while introducing new observational information. In implementation, this is typically approximated by a parameterized function

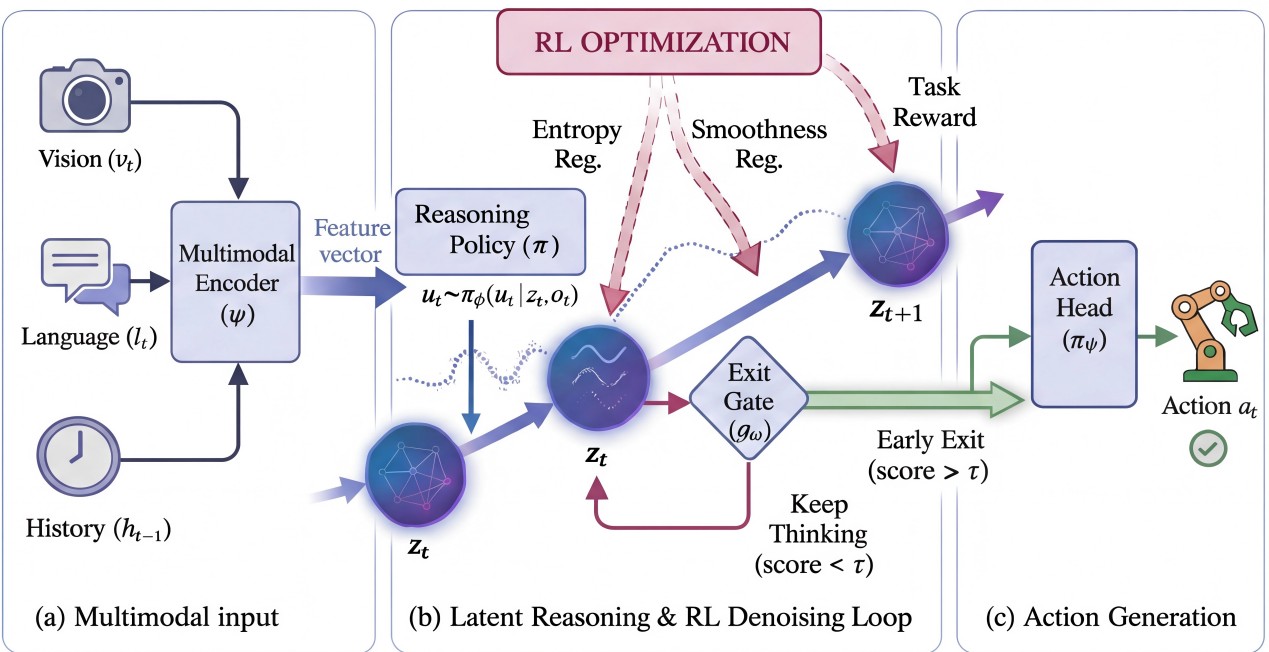

*Figure 2.* **The overall framework of *AVA-VLA*.** The architecture operates in three stages: **(a) Multimodal Encoding:** Visual ($v_t$), linguistic ($l_t$), and historical ($h_{t-1}$) inputs are encoded into a unified feature vector. **(b) Latent Reasoning & RL Denoising Loop:** The core reasoning process is modeled as a POMDP. A *Reasoning Policy* $\pi_\phi$ generates an internal update action $u_t$ to evolve the latent state $z_t$. This process is optimized via Reinforcement Learning (RL) with entropy and smoothness regularization to ensure stability. An *Exit Gate* $g_\omega$ dynamically evaluates state confidence. **(c) Action Generation:** Once the gate triggers an exit (score $> \tau$), the finalized latent state is projected by the Action Head $\pi_\psi$ to execute the robotic action $a_t$.

to model the continuous temporal evolution of the latent state.

- **Reward Function** ($R$)**:** The function $R(z_t, a_t)$ measures the performance of the latent reasoning state and the guided action decision at the task level. This reward can originate from direct environmental feedback on action outcomes or include regularization constraints on reasoning stability and consistency, ensuring the optimization of the latent state aligns with the final task objective.

Through this modeling, the reasoning process is no longer viewed as a passively generated intermediate representation but is explicitly characterized as a dynamic decision process controlled by a reasoning policy. This POMDP formulation lays the foundation for treating latent reasoning generation as a sequential decision problem and utilizing reinforcement learning for denoising and stabilization.

### 3.3. Reasoning Policy and Latent State Update

To operationalize the reasoning POMDP, we instantiate the abstract transition dynamics with parameterized neural networks, focusing on policy definition and state evolution.

**Definition of Reasoning Policy.** To model the latent reasoning process as a controllable and optimizable decision

process, we introduce a parameterized reasoning policy $\pi_\phi$. This policy generates a *latent reasoning update action* given the current latent reasoning state and multimodal observation:

$$u_t \sim \pi_\phi(u_t \mid z_t, o_t), \tag{3}$$

where $z_t \in \mathcal{Z}$ denotes the latent reasoning state at time step $t$, $o_t \in \mathcal{O}$ represents the multimodal observation, and $u_t \in \mathcal{U}$ is the reasoning update action. In the main experiments, $\mathcal{U} \subset \mathbb{R}^{64}$ is continuous and $\pi_\phi$ is parameterized as a diagonal Gaussian:

$$\pi_\phi(u_t \mid z_t, o_t) = \mathcal{N}\left(u_t; \mu_\phi(z_t, \tilde{o}_t), \mathrm{diag}(\sigma_\phi^2(z_t, \tilde{o}_t))\right), \tag{4}$$

where $\tilde{o}_t = \psi(o_t)$ is the encoded multimodal observation, and $\mu_\phi(\cdot)$ and $\sigma_\phi(\cdot)$ are produced by the reasoning policy network. We found this continuous modulation to provide smoother gradients and more stable PPO optimization than a discrete update selector; a discrete Softmax policy is a valid alternative but is not used for the main results.

In this design, the update action $u_t$ can be interpreted as a selection of the update mode or a modulation signal (e.g., controlling the intensity of information injection, attention distribution, or update direction). By explicitly modeling the reasoning policy, the model can adaptively determine the reasoning update mechanism based on the current state

and observation at each time step, rather than relying on fixed, task-agnostic state update rules.

**Evolution of Latent Reasoning State.** Once the reasoning policy generates an update action, the latent reasoning state updates according to a parameterized state transition function $f_\theta$. The evolution process is expressed as:

$$z_{t+1} = f_\theta(z_t, o_t, u_t), \tag{5}$$

where $f_\theta$ is the latent state transition function with parameters $\theta$. To explicitly characterize the update process, we expand this formulation as:

$$\begin{aligned} \tilde{o}_t &= \psi(o_t), \\ \Delta z_t &= g_\theta(z_t, \tilde{o}_t, u_t), \\ z_{t+1} &= z_t + \Delta z_t, \end{aligned} \tag{6}$$

where $\psi(\cdot)$ is the multimodal encoder, and $g_\theta(\cdot)$ is the reasoning update network responsible for generating the incremental update $\Delta z_t$ for the current state. This incremental form helps maintain the continuity and stability of the reasoning state over the temporal dimension.

In specific implementations, $g_\theta$ can be realized via Recurrent Neural Networks (RNNs), Transformer Blocks, or Gated structures. For instance:

$$\Delta z_t = \alpha(u_t) \odot \text{Transformer}_\theta(z_t, \tilde{o}_t), \tag{7}$$

where $\alpha(u_t)$ represents gating coefficients controlled by the reasoning update action $u_t$, and $\odot$ denotes element-wise multiplication. In this way, the reasoning update action directly regulates the magnitude and direction of the latent state update.

This mechanism ensures that the update of the reasoning state depends not only on observational inputs but is also subject to explicit control by the policy level, transforming the generation of latent reasoning into a learnable and adjustable dynamic system. This modeling approach provides the necessary structural foundation for subsequently applying regularization constraints, introducing reasoning denoising mechanisms, and implementing adaptive reasoning depth control.

### 3.4. Coupling Action Policy with Latent Reasoning

To enable the latent reasoning state to directly serve the downstream decision-making process, we introduce an action policy $\pi_\psi$ conditioned on the latent reasoning state. This policy generates the environmental action given the current latent state and multimodal observation:

$$a_t \sim \pi_\psi(a_t \mid z_t, o_t), \tag{8}$$

where $a_t \in \mathcal{A}$ represents the environmental action at time step $t$, and $\psi$ denotes the parameters of the action policy.

Through this design, the latent reasoning state $z_t$ is explicitly positioned at the core of the decision process. It serves as a compressed representation and task-relevant abstraction of the multimodal observation, filtering out redundancy and noise from the raw inputs, thereby enhancing decision stability and consistency.

In implementation, the action policy is parameterized as a conditional probability distribution, expressible as:

$$\pi_\psi(a_t \mid z_t, o_t) = \text{Softmax}\big(h_\psi([z_t, \psi(o_t)])\big), \tag{9}$$

where $h_\psi(\cdot)$ is the action policy network, $\psi(o_t)$ is the encoded observation, and $[\cdot, \cdot]$ denotes feature concatenation. This structure allows the action generation process to be explicitly conditioned on the latent reasoning state while retaining responsiveness to current observational information.

Under this joint reasoning and decision modeling, the complete interaction trajectory:

$$\tau = \{(z_t, o_t, u_t, a_t)\}_{t=1}^T \tag{10}$$

is jointly generated by the reasoning policy and the action policy. The dynamic process can be summarized as:

$$\begin{aligned} u_t &\sim \pi_\phi(u_t \mid z_t, o_t), \\ z_{t+1} &= f_\theta(z_t, o_t, u_t), \\ a_t &\sim \pi_\psi(a_t \mid z_t, o_t). \end{aligned} \tag{11}$$

Here, the reasoning policy $\pi_\phi$ controls the mode of latent state updates, while the action policy $\pi_\psi$ executes environmental interactions based on the current latent state, forming a tight coupling in the temporal dimension.

The overall learning objective is defined to maximize the expected cumulative reward:

$$\max_{\phi, \theta, \psi} \ \mathbb{E}_{\tau \sim (\pi_\phi, \pi_\psi)} \left[ \sum_{t=1}^T \gamma^t r(z_t, a_t) \right], \tag{12}$$

where $r(z_t, a_t)$ represents the immediate reward function associated with the latent reasoning state and the guided action, and $\gamma \in (0, 1]$ is the discount factor. This optimization objective encourages the model to learn a joint reasoning-decision strategy that not only forms high-quality latent reasoning representations but also effectively guides action decisions.

Through this joint modeling and optimization, the latent reasoning state ceases to exist merely as an intermediate variable. Instead, via direct optimization of long-term rewards, it achieves end-to-end alignment with task goals and decision performance, realizing a tight synergy between the reasoning process and action decision-making.

### 3.5. RL-based Latent Reasoning Denoising

**Motivation.** Since the latent reasoning state $z_t$ lacks explicit supervision, its generation relies entirely on the implicit parameterization of the reasoning policy and transition dynamics. In scenarios with noisy multimodal inputs, stochastic initialization, or long-horizon sequences, the latent state is prone to representation drift and instability due to error accumulation. This instability propagates to the action policy, ultimately degrading decision performance. To mitigate this, we treat latent reasoning generation as an optimizable sequential decision process. By introducing Reinforcement Learning (RL), we impose task-oriented constraints that align the evolution of $z_t$ directly with long-term task rewards, thereby enhancing the stability and discriminability of the reasoning trajectory without requiring external annotations.

**Reward Function Design.** To impose effective constraints, we define the immediate reward function $r_t$ for the reasoning policy as follows:

$$r_t = r_{\text{task}}(a_t) - \lambda_1 \mathcal{H}\big(\pi_\phi(\cdot \mid z_t, o_t)\big) - \lambda_2 \|z_{t+1} - z_t\|^2, \tag{13}$$

where $r_{\text{task}}(a_t)$ denotes the task-level reward (e.g., success signal) measuring decision quality. The second term is an entropy regularization term, designed to penalize excessive uncertainty in the reasoning update distribution, thereby reducing stochastic perturbations. The third term is a smoothness regularization term that penalizes the magnitude of changes between consecutive latent states, encouraging temporal continuity and stability. This composite reward jointly constrains the latent state from three dimensions: task performance, randomness control, and representation stability. Crucially, the smoothness term does not force the state to remain static; rather, it suppresses irrelevant updates caused by noise while retaining responsiveness to significant, task-critical information changes.

**Reasoning Policy Optimization.** Under this reward formulation, we employ the Actor-Critic method to optimize the reasoning policy $\pi_\phi$. We define the state value function based on the latent state:

$$V^\pi(z_t) = \mathbb{E}_{\tau \sim \pi} \left[ \sum_{k=t}^{T} \gamma^{k-t} r_k \right], \tag{14}$$

where the expectation is over trajectories generated jointly by the reasoning and action policies. The optimization objective $J(\phi)$ is to maximize the expected cumulative reward:

$$J(\phi) = \mathbb{E}_{\tau \sim \pi} \left[ \sum_{t=1}^{T} \gamma^t r_t \right]. \tag{15}$$

The gradient is approximated via the Policy Gradient Theorem:

$$\nabla_\phi J(\phi) \approx \mathbb{E} \left[ \nabla_\phi \log \pi_\phi(u_t \mid z_t, o_t) \left( R_t - V^\pi(z_t) \right) \right], \tag{16}$$

where $R_t = \sum_{k=t}^{T} \gamma^{k-t} r_k$ is the discounted return. By introducing the learned value function $V^\pi(z_t)$ as a baseline, this update effectively reduces gradient variance and stabilizes optimization. This process drives the policy to generate trajectories that are both task-aligned and temporally coherent, achieving robust reasoning denoising via long-term coupling with action decisions. In implementation, we use PPO with generalized advantage estimation (GAE) to propagate sparse task success signals back to each latent update. Thus the RL stage assigns credit to intermediate reasoning actions through the learned critic rather than relying only on the final success indicator.

### 3.6. Adaptive Reasoning Depth and Early-Exit

**Motivation.** In multimodal decision-making, the requirement for reasoning depth varies significantly across time steps and task states. In many instances, the model can form a sufficient understanding of the task strategy at a shallow reasoning stage; continuing updates would incur computational costs and potentially dilute state discriminability due to noise accumulation. Furthermore, in long sequences, reasoning updates do not monotonically improve performance—once a high-confidence state is reached, further updates may induce unnecessary oscillation. Therefore, introducing a mechanism to adaptively terminate reasoning when it is "sufficiently good" is critical for trading off decision performance and inference efficiency.

**Exit Determination Function.** To implement adaptive termination, we introduce a parameterized Exit Determination Function:

$$e_t = g_\omega(z_t), \tag{17}$$

where $g_\omega(\cdot)$ is the determination network parameterized by $\omega$. It takes the current latent state $z_t$ as input and outputs a scalar $e_t \in \mathbb{R}$, representing the estimated confidence or sufficiency of the current state for decision-making. During inference, the reasoning process terminates if the output satisfies:

$$e_t > \tau, \tag{18}$$

where $\tau$ is a calibrated threshold. Upon termination, the model stops generating update actions and directly executes the action decision:

$$a_t \sim \pi_\psi(a_t \mid z_t, o_t). \tag{19}$$

If the condition is not met, the reasoning update continues. This Early-Exit mechanism introduces a state-dependent adaptive control signal without altering the fundamental

modeling, effectively pruning redundant computation when the reasoning state stabilizes. After policy training, $g_\omega$ is calibrated with binary labels indicating whether additional latent reasoning yields less than a small marginal improvement. We use $\tau = 0.55$ in the main experiments, selected by the threshold-sensitivity sweep in Table 5.

# 4. Experiments

## 4.1. Datasets and Tasks

The **LIBERO benchmark** (Table 1) is designed for lifelong robot learning, utilizing a Franka Emika Panda arm and running in the MuJoCo physics engine. This benchmark includes four distinct test suites: **LIBERO-Spatial**, **LIBERO-Object**, **LIBERO-Goal**, and **LIBERO-Long**, each testing the robot's performance in spatial reasoning, object manipulation, goal achievement, and long-horizon tasks. The benchmark consists of 5,000 episodes and 100 tasks, with data including RGB images, proprioceptive states, and delta actions. All tasks are procedurally generated to ensure diversity in object configurations, goals, and environmental conditions. Additionally, we extend the evaluation to **LIBERO+**, a more complex benchmark that introduces 7 perturbation dimensions and 21 sub-dimensions, designed to systematically assess the model's performance across various challenges.

The **CALVIN benchmark** (Table 3) focuses on long-horizon robot manipulation conditioned on language, using a Franka Panda arm with RGBD observations and proprioception. This benchmark evaluates sequential reasoning capabilities under multimodal inputs, with a total of 34 tasks spanning four different environments (A-D), and more than 20,000 episodes. The tasks require the robot to perform multi-step operations such as "open the drawer, pick up the blue block, push it into the drawer," and emphasize the model's ability to generalize to **unseen objects**. We evaluate using the **ABC→D** setting, where the model is trained on environments A, B, and C and tested on environment D to assess the model's generalization to unseen environments.

## 4.2. Implementation Details

Hyperparameters include a visual feature dimension of 768 and a language embedding dimension of 768. The reasoning and action networks use hidden layer sizes of 1024 and 2048, with a dropout rate of 0.1. The model is optimized with Adam ($1 \times 10^{-4}$ learning rate, $\beta_1 = 0.9$, $\beta_2 = 0.999$, $\epsilon = 10^{-8}$) and a batch size of 32. Training follows three stages: 100K behavior-cloning pretraining steps, 50K latent-reasoning warmup steps, and joint PPO fine-tuning. The RL stage uses $\gamma = 0.99$, GAE with $\lambda = 0.95$, PPO clip ratio 0.2, gradient clipping at 1.0, and entropy and smoothness regularizers to stabilize latent reasoning. The PPO stage

uses about 1.2M environment interaction steps and takes 18.6 hours on 8 NVIDIA A100 80GB GPUs. The early-exit threshold is set to $\tau = 0.55$ based on validation calibration.

## 4.3. Results

First, on the **LIBERO** benchmark, the **Ours** method demonstrates significant superiority across multiple task suites. In the setup where a single policy is applied to all four task suites, Ours outperforms other methods, particularly in **Object SR** (99.4%), **Goal SR** (97.8%), and **Long SR** (98.1%), achieving an overall average success rate of **98.3%**, which is the highest among all methods. Even when employing a separate policy per task suite, Ours continues to exhibit superior performance, especially in **Spatial SR** (99.6%) and **Object SR** (99.7%), surpassing other approaches. Overall, **Ours** shows remarkable multi-task adaptability and stability in both experimental settings, demonstrating its robustness in a variety of scenarios.

In the analysis of the **CALVIN ABC→D** benchmark, **Ours** also performs exceptionally well, particularly in terms of task completion steps and average task length. Ours leads in the number of steps required to complete tasks, achieving an **84.0%** success rate for five-step tasks, which is significantly higher than other methods such as **FLOWER** (77.8%) and **VLA-Adapter** (76.5%). This indicates that Ours excels in longer tasks, showing greater stability and efficiency in completing multi-step procedures. Furthermore, Ours also excels in **average task length**, achieving **4.77**, which reflects its strength in executing extended tasks with high success rates.

## 4.4. Computational Efficiency Analysis

We provide detailed latency measurements on an NVIDIA A100 GPU with batch size 1 in Table 2.

While AVA-VLA incurs a marginal latency overhead compared to pure reflex-based architectures like $\pi_0$-FAST (145ms vs. 98ms), this additional computation represents a strategic allocation of inference budget to *latent reasoning*. This investment yields substantial returns in decision stability, improving long-horizon success rates by **4.1%** on LIBERO-Long compared to the fastest baseline.

Crucially, compared to Explicit CoT methods (892ms), our method achieves a **6×** **speedup**, effectively bridging the gap between high-level reasoning and real-time control. The Early-Exit mechanism dynamically optimizes this trade-off, reducing average reasoning depth by 54% (from 5.0 to 2.3 steps) with minimal performance degradation. We further evaluate the exit threshold in Table 5; the sweep shows a smooth latency-performance frontier rather than brittle task-specific tuning. Under a matched latency budget, AVA-VLA also outperforms explicit CoT baselines because compact

*Table 1.* Comparison on the LIBERO benchmark. The results are reported in two groups: one policy for all 4 suites, and one policy per suite. The best results in each column of each group are highlighted in **bold**.

| Method | Spatial SR (%) | Object SR (%) | Goal SR (%) | Long SR (%) | Average SR (%) |
|---|---|---|---|---|---|
| *One policy for all 4 suites* | | | | | |
| TraceVLA (Zheng et al., 2024) | 84.6 | 85.2 | 75.1 | 54.1 | 74.8 |
| WorldVLA (Cen et al., 2025) | 87.6 | 96.2 | 83.4 | 60.0 | 81.8 |
| $\pi_0$ (Black et al., 2024) | 96.8 | 98.8 | 95.8 | 85.2 | 94.2 |
| $\pi_0$-FAST (Pertsch et al., 2025) | 96.4 | 96.8 | 88.6 | 60.2 | 85.5 |
| UnifiedVLA (Wang et al., 2025b) | 95.4 | 98.8 | 93.6 | 94.0 | 95.5 |
| OpenVLA-OFT (Kim et al., 2025) | 97.7 | 98.0 | 96.1 | 95.3 | 96.8 |
| Ours | **97.8** | **99.4** | **97.8** | **98.1** | **98.3** |
| *One policy per suite* | | | | | |
| OpenVLA (Kim et al., 2024) | 84.7 | 88.4 | 79.2 | 53.7 | 76.5 |
| SpatialVLA (Qu et al., 2025) | 88.2 | 89.9 | 78.6 | 55.5 | 78.1 |
| CoT-VLA (Zhao et al., 2025) | 87.5 | 91.6 | 87.6 | 69.0 | 83.9 |
| NORA (Hung et al., 2025) | 92.2 | 95.4 | 89.4 | 74.6 | 87.9 |
| PD-VLA (Song et al., 2025) | 95.5 | 96.7 | 94.9 | 91.7 | 94.7 |
| UniVLA (Bu et al., 2025) | 96.5 | 96.8 | 95.6 | 92.0 | 95.2 |
| OpenVLA-OFT (Kim et al., 2025) | 97.6 | 98.4 | 97.9 | 94.5 | 97.1 |
| FLOWER (Reuss et al., 2025) | 97.5 | 99.1 | 96.1 | 94.9 | 96.9 |
| VLA-Adapter (Wang et al., 2025a) | 97.8 | 99.2 | 97.2 | 95.0 | 97.3 |
| RIPT-VLA (Tan et al., 2025) | 99.0 | 98.6 | 98.6 | 93.8 | 97.5 |
| Ours | **99.6** | **99.7** | **98.7** | **96.5** | **98.6** |

*Table 2.* Latency comparison on LIBERO-Spatial. "Avg. Steps" for our method refers to the number of latent reasoning iterations.

| Method | LIBERO Avg. Steps | Mean Latency (ms) | P90 Latency (ms) | Throughput (Hz) |
|---|---|---|---|---|
| OpenVLA(Kim et al., 2024) | 1.0 | 127 | 135 | 7.9 |
| CoT-VLA(Zhao et al., 2025) | 8.5 | 892 | 1,240 | 1.1 |
| $\pi_0$-FAST(Pertsch et al., 2025) | 1.0 | 98 | 102 | 10.2 |
| PD-VLA(Song et al., 2025) | 1.0 | 76 | 82 | 13.2 |
| Ours (w/o Early-Exit) | 5.0 | 312 | 340 | 3.2 |
| **Ours (Full)** | **2.3** | **145** | **189** | **6.9** |

*Table 3.* Comparison on the CALVIN ABC→D benchmark in terms of success rates (%) and average length. The best results in each column are highlighted in **bold**.

| CALVIN ABC→D | Task completed in a row ↑ | | | | | Avg. len ↑ |
|---|---|---|---|---|---|---|
| | 1 | 2 | 3 | 4 | 5 | |
| OpenVLA (Kim et al., 2024) | 91.3 | 77.8 | 62.0 | 52.1 | 43.5 | 3.27 |
| UniVLA (Bu et al., 2025) | 95.5 | 85.8 | 75.4 | 66.9 | 56.5 | 3.80 |
| UnifiedVLA (Wang et al., 2025b) | 98.9 | 94.8 | 89.0 | 82.8 | 75.1 | 4.41 |
| OpenVLA-OFT (Kim et al., 2025) | 96.9 | 92.0 | 85.7 | 80.4 | 72.9 | 4.28 |
| FLOWER (Reuss et al., 2025) | 99.4 | 95.8 | 90.7 | 84.9 | 77.8 | 4.53 |
| VLA-Adapter (Wang et al., 2025a) | 99.1 | 94.6 | 88.8 | 82.8 | 76.5 | 4.42 |
| Seer (Tian et al., 2024) | 96.3 | 91.6 | 86.1 | 80.3 | 74.0 | 4.28 |
| Ours | **99.7** | **96.5** | **94.5** | **91.1** | **84.0** | **4.77** |

*Table 4.* Ablation study on LIBERO (Avg. SR) and CALVIN (Avg. Len).

| METHOD VARIANT | LATENT REAS. | LIBERO SR (%) | CALVIN AVG. LEN |
|---|---|---|---|
| OPENVLA-OFT | - | 97.1 | 4.28 |
| W/O LATENT REASONING | ✗ | 95.8 | 4.05 |
| W/O RL DENOISING | ✓ | 96.6 | 4.21 |
| W/O LATENT SMOOTHNESS | ✓ | 96.9 | 4.33 |
| W/O EARLY-EXIT | ✓ | 98.0 | 4.61 |
| FULL | ✓ | **98.3** | **4.65** |

latent updates avoid token-by-token decoding.

### 4.5. Ablation Study

We analyze the contribution of each component in Table 4. Removing the *Latent Reasoning* module causes the most significant drop (LIBERO SR: 98.3% → 95.8%), validating the importance of implicit intermediate abstraction. Removing *RL Denoising* also degrades performance (96.6%),

confirming that unsupervised latent states are prone to noise without task-driven constraints. The *Early-Exit* mechanism, when removed, maintains high success rates but slightly reduces long-horizon stability (CALVIN len: 4.65 → 4.61) and increases computational cost.

To separate latent reasoning from mere capacity increase, Appendix C.2 reports a capacity-matched behavior-cloning

*Table 5.* Early-exit threshold sensitivity on LIBERO.

| $\tau$ | Avg. Steps | Latency (ms) | Avg. SR (%) |
|---|---|---|---|
| 0.30 | 1.5 | 108 | 96.1 |
| 0.40 | 1.8 | 121 | 97.0 |
| 0.50 | 2.1 | 136 | 97.8 |
| **0.55** | **2.3** | **145** | **98.3** |
| 0.65 | 2.8 | 167 | 98.2 |
| 0.75 | 3.4 | 196 | 98.2 |
| 0.85 | 4.1 | 234 | 98.1 |
| 0.95 | 4.7 | 278 | 98.0 |
| 1.00 (no exit) | 5.0 | 312 | 98.0 |

*Table 6.* Latent-state distance statistics on LIBERO. Near-zero ratio measures the fraction of steps with $\|z_{t+1} - z_t\|^2 < 10^{-3}$.

| Method | Mean | Med. | P95 | Near-zero |
|---|---|---|---|---|
| w/o RL | 0.11 | 0.08 | 0.29 | 18.40% |
| AVA-VLA | 0.18 | 0.14 | 0.44 | 7.20% |
| w/o Smooth. | 0.36 | 0.28 | 0.91 | 2.40% |

latent baseline with the same backbone and latent depth but without the POMDP formulation or RL denoising. Appendix C.3 compares against explicit CoT under a matched inference budget. Appendix C.4 further probes frozen latent states and shows that they encode end-effector geometry, gripper state, and subgoal progress, with RL denoising improving both task performance and latent-state readability.

Table 6 addresses the state-collapse concern directly. AVA-VLA keeps a nonzero latent-state change rate under smoothness regularization, reducing the near-zero ratio relative to the no-RL variant while avoiding the overly large latent jumps observed without the smoothness term.

## 5. Conclusion

This paper introduces the Adaptive Variable Alignment Visual-Language-Action Model (AVA-VLA), a novel framework that redefines reasoning as an implicit, task-aligned latent variable evolution, instead of relying on explicit text generation. By modeling inference as a partially observable Markov decision process (POMDP) and using reinforcement learning, we optimize the reasoning process with a composite reward that balances task success, entropy regularization, and temporal smoothness. The early termination mechanism adapts to balance computational efficiency and decision performance. Extensive experiments on LIBERO and CALVIN benchmarks show that AVA-VLA significantly reduces inference latency while improving success rate and long-horizon task stability, offering a new approach for efficient and robust embodied intelligence systems.

## Acknowledgements

This work is supported by Beijing Natural Science Foundation under Grant No. QY25048.

## Impact Statement

This work introduces AVA-VLA to improve the energy efficiency of robotic control via an "Early-Exit" reasoning mechanism. While prioritizing speed can present stability challenges in physical environments, our framework mitigates this by aligning the reasoning process with task-success rewards through reinforcement learning and by using conservative exit calibration under uncertainty. We also advise practitioners to be mindful of potential biases inherited from the pre-trained vision-language backbones used in this system, and to deploy physical robots with external safety monitors and task-specific validation.

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

# A. Idealized Objective-Level Analysis

### A.1. Idealized View of RL-based Denoising

The practical AVA-VLA model is a large nonlinear architecture, so we do not claim a global convergence guarantee for the coupled reasoning-action optimization. Instead, the following analysis explains the intended local effect of the denoising objective. The reward combines task success, entropy control, and smoothness:

$$r_t = r_{\text{task}}(a_t) - \lambda_1 \mathcal{H}(\pi_\phi(\cdot \mid z_t, o_t)) - \lambda_2 \|z_{t+1} - z_t\|^2. \tag{20}$$

Along trajectories visited during training, the smoothness term discourages high-frequency latent perturbations, while the task reward preserves updates that improve downstream control. This gives a local denoising bias: irrelevant fluctuations are suppressed, but task-critical state transitions can still produce nonzero changes in $z_t$. The latent-distance diagnostics in Appendix C.4 empirically support this view by showing that $\|z_{t+1} - z_t\|^2$ does not collapse to zero and instead peaks near semantic phase changes such as grasp-to-move and move-to-place transitions.

### A.2. Early-Exit as Approximate Stopping

Early exit can be interpreted as an approximate value-of-computation decision. Let $c > 0$ denote the computational cost of one more latent reasoning step and let $V(z_t)$ denote the expected downstream value from the current latent state. An ideal stopping rule would stop when the expected marginal improvement from another update is smaller than the computation cost:

$$\tau^* = \min \{t : \mathbb{E}[V(z_t) - V(z_{t+1})] < c\} \tag{21}$$

The learned gate $g_\omega$ is a lightweight approximation to this criterion. It is calibrated with labels indicating whether additional reasoning brings only a small marginal improvement, and the empirical threshold sweep in Appendix C.1 shows that this approximation yields a stable latency-performance trade-off.

# B. Implementation Details

### B.1. Implementation Details

The main experiments are based on the OpenVLA-OFT architecture, which integrates a `SigLIP-DINOv2` backbone for visual feature extraction, a 3-layer MLP with GELU activation to map visual features to the language space, and a 2-layer MLP with GELU activation to project proprioceptive states into the same space. Continuous actions are generated via a 4-layer MLP with ReLU activation. Unlike standard OpenVLA, bidirectional attention is used instead of causal attention, enabling parallel decoding and chunked action outputs at each timestep, reducing inference time. The architecture uses the `Llama-2 7B` language model for multimodal processing.

The system processes multimodal observations (visual, language, proprioceptive) into a latent reasoning state that evolves through latent variables and guides action generation. By optimizing the reasoning process with RL and using early-exit, the architecture balances performance with computational efficiency. Evaluations on LIBERO and CALVIN benchmarks demonstrate superior efficiency and stability compared to other methods.

### B.2. Network Architectures

**Multimodal Encoder $\psi(\cdot)$.** Visual features are extracted using the concatenated output of SigLIP (ViT-SO400M-14-SigLIP-384) and DINOv2 (ViT-L/14). The visual encoder produces 768-dimensional features, projected to match the language model's embedding space via a 3-layer MLP.

Language instructions are tokenized using Llama-2's tokenizer and encoded via the frozen Llama-2 7B backbone. Proprioceptive states (7-DoF joint positions, gripper state) are projected using a 2-layer MLP with GELU activation.

**Reasoning Policy Network $\pi_\phi$.** The reasoning policy is implemented as a continuous Gaussian policy over latent update actions. In its general continuous formulation, the reasoning policy $\pi_\phi(u_t \mid z_t, o_t)$ is parameterized as:

$$\pi_\phi(u_t \mid z_t, o_t) = \mathcal{N}(u_t; \mu_\phi(z_t, \tilde{o}_t), \text{diag}(\sigma_\phi^2(z_t, \tilde{o}_t))) \tag{22}$$

where $\mu_\phi$ and $\sigma_\phi$ are outputs of a 4-layer Transformer with 8 attention heads, hidden dimension 512, and ReLU activations. The update action space $\mathcal{U} \subset \mathbb{R}^{64}$ represents continuous modulation signals.

**State Transition Function** $f_\theta$. We implement $g_\theta$ using a Gated Recurrent Unit (GRU) with hidden dimension 1024, augmented with cross-attention to multimodal observations:

$$\Delta z_t = \text{GRU}_\theta([z_t; \tilde{o}_t], u_t) \odot \sigma(\text{MLP}_\alpha(u_t)) \tag{23}$$

where $\sigma(\cdot)$ is the sigmoid function implementing the gating mechanism described in Section 3.3.

**Exit Gate** $g_\omega$. The confidence estimator is a lightweight 2-layer MLP with residual connections:

$$e_t = \text{Sigmoid}(\text{MLP}_\omega(z_t)) \in (0, 1) \tag{24}$$

### B.3. Training Procedure

**Stage 1: Behavior Cloning Pretraining.** We first pretrain the multimodal encoder and action policy using standard behavior cloning on the training datasets for 100K steps with batch size 64. The reasoning policy is initialized to output zero-mean Gaussian noise.

**Stage 2: Latent Reasoning Warmup.** We freeze the action policy and train the reasoning policy and transition function for 50K steps using only the smoothness regularization (setting $\lambda_1 = 0$ in the reward). This establishes stable latent dynamics before introducing RL optimization.

**Stage 3: Joint RL Fine-tuning.** We jointly optimize all components using PPO (Schulman et al., 2017) with the hyperparameters in Table 7. Sparse task rewards are propagated through GAE and a learned value function, assigning credit to each latent update action. The PPO stage uses about 1.2M environment interaction steps and takes about 18.6 hours on 8 NVIDIA A100 80GB GPUs.

*Table 7.* Hyperparameters for PPO training.

| Hyperparameter | Value |
| --- | --- |
| Learning rate (policy) | $3 \times 10^{-5}$ |
| Learning rate (critic) | $1 \times 10^{-4}$ |
| PPO clip ratio $\epsilon$ | 0.2 |
| GAE parameter $\lambda$ | 0.95 |
| Entropy coefficient $\lambda_1$ | 0.01 |
| Smoothness coefficient $\lambda_2$ | 0.1 |
| Batch size | 512 |
| Mini-batch size | 64 |
| PPO epochs | 4 |
| Gradient clipping | 1.0 |

**Stage 4: Exit Gate Calibration.** After policy convergence, we train $g_\omega$ using a binary cross-entropy loss where positive labels are assigned to states where continuing reasoning for $k$ additional steps (we use $k = 3$) yields marginal improvement less than threshold $\delta = 0.05$.

### B.4. Inference Optimization

During inference, we implement dynamic batching for the reasoning steps. When the exit gate triggers for a subset of samples in a batch, those samples immediately proceed to action generation while others continue reasoning. This requires careful memory management but achieves up to 40% latency reduction compared to synchronous execution.

### B.5. Reproducibility Checklist

All reported main results use the same OpenVLA-OFT backbone, the same multimodal encoder, and the same action head unless otherwise stated. The PPO effective batch size is 512, the mini-batch size is 64, and each update uses 4 PPO epochs. We evaluate stability over three random seeds in Table 8.

*Table 8.* Stability across random seeds.

| Seed | LIBERO Avg. SR (%) | CALVIN Avg. Len | Mean Latency (ms) |
|---|---|---|---|
| 0 | 98.1 | 4.59 | 148 |
| 1 | 98.3 | 4.65 | 145 |
| 2 | 98.5 | 4.71 | 143 |
| Mean $\pm$ Std | 98.3 $\pm$ 0.20 | 4.65 $\pm$ 0.06 | 145.3 $\pm$ 2.1 |

# C. Additional Experiments

## C.1. Exit Threshold Sensitivity

Table 5 reports the early-exit threshold sweep. The results show a smooth latency-performance trade-off rather than brittle threshold sensitivity. The setting $\tau = 0.55$ gives the best balance in our sweep, reaching 98.3% average LIBERO success with 2.3 reasoning steps and 145 ms mean latency. Higher thresholds allocate more computation but do not improve success, while very low thresholds exit prematurely and reduce performance.

## C.2. Capacity-Matched Latent Baseline

To test whether the performance gain comes merely from a stronger recurrent controller or more parameters, we compare against a capacity-matched behavior-cloning latent baseline. This baseline uses the same backbone and the same latent depth as AVA-VLA, but removes the POMDP formulation and RL denoising. It is intended as a capacity-matched no-RL control rather than a full structure-replaced Iso-FLOPs proof. Table 9 shows that simply adding a latent module does not explain the gain; the full model improves because latent updates are treated as policy-controlled actions and optimized with task-aligned RL.

*Table 9.* Capacity-matched comparison.

| Method | Same Backbone | Same Latent Depth | POMDP | RL Denoising | LIBERO Avg. SR (%) | CALVIN Avg. Len |
|---|---|---|---|---|---|---|
| OpenVLA-OFT | ✓ | ✗ | ✗ | ✗ | 97.1 | 4.28 |
| Controlled BC Latent Module | ✓ | ✓ | ✗ | ✗ | 96.6 | 4.21 |
| AVA-VLA (Full) | ✓ | ✓ | ✓ | ✓ | **98.3** | **4.65** |

## C.3. Compute-Matched Explicit CoT Comparison

We also compare AVA-VLA against explicit CoT under a controlled inference budget. The explicit-CoT baseline is constrained to the same latency range as AVA-VLA, which forces it to truncate autoregressive reasoning. As shown in Table 10, latent parallel reasoning retains high task success under this budget, while truncated token-level CoT suffers a large accuracy loss.

*Table 10.* Comparison under matched inference latency.

| Method | Reasoning Form | Mean Latency (ms) | Success Rate (%) |
|---|---|---|---|
| Behavior Cloning | No reasoning | 92 | 91.2 |
| Explicit CoT | Autoregressive text | 143 | 78.8 |
| AVA-VLA | Parallel latent reasoning | 144 | **98.3** |

## C.4. Latent-State Diagnostics

Latent reasoning is less directly interpretable than natural-language CoT, so we diagnose what the frozen latent states encode. We train lightweight probes to predict end-effector position, gripper state, and subgoal progress from $z_t$. Table 11 shows that AVA-VLA latent states encode structured information about geometry, execution state, and task progress. The gain over the variant without RL denoising indicates that denoising improves not only final success but also the stability and readability of the latent trajectory.

*Table 11.* Probing frozen latent states.

| Method | End-effector MAE ↓ | Gripper Acc. ↑ | Subgoal Progress $R^2$ ↑ |
|---|---|---|---|
| w/o Latent Reasoning | 2.71 cm | 89.1 | 0.58 |
| w/o RL Denoising | 1.94 cm | 92.7 | 0.73 |
| AVA-VLA (Full) | **1.39 cm** | **96.2** | **0.84** |

We also examine the latent feature distance $\|z_{t+1} - z_t\|^2$ across RL fine-tuning trajectories. Table 6 reports quantitative statistics on LIBERO. The distance does not vanish under the smoothness term: AVA-VLA reduces the near-zero ratio compared with the no-RL variant while avoiding the overly large latent jumps observed without smoothness. This suggests that smoothness regularization suppresses noisy oscillations without forcing latent-state collapse. Qualitatively, the distance also exhibits meaningful peaks near semantic transitions, such as grasp-to-move and move-to-place phases.

### C.5. Robustness Analysis on LIBERO+

LIBERO+ introduces systematic perturbations including visual distractions (lighting, texture), physical parameter variations (mass, friction), and observation noise. Results are in Table 12.

**Mechanisms of Robustness.** As shown in Table 12, AVA-VLA demonstrates superior resilience across two key dimensions:

- **Visual Stability (Lighting & Texture):** Our method achieves significant gains under visual shifts (e.g., **+9.1%** over OpenVLA-OFT in Lighting conditions). This result validates that the *Latent Reasoning* module acts as an effective information bottleneck, filtering out high-frequency visual noise and retaining semantic task features before they reach the action generation head.

- **Dynamics Robustness (Friction & Mass):** The ablation study highlights the critical role of the *RL Denoising* mechanism. Comparing "Ours (w/o RL Denoising)" to "Ours (Full)", we observe a substantial performance gap in Object Mass (**+3.8%**) and Joint Friction (**+4.6%**). This suggests that without the RL constraints (specifically smoothness and task-reward alignment), the latent state fails to adapt to subtle changes in physical dynamics. The RL optimization effectively forces the latent trajectory to encode "physics-compliant" reasoning that generalizes better across dynamics perturbations.

*Table 12.* Success rates (%) under LIBERO+ perturbations.

| Perturbation Type | OpenVLA-OFT | Ours (w/o RL Denoising) | **Ours (Full)** |
|---|---|---|---|
| Lighting | 82.3 | 85.1 | **91.4** |
| Texture | 78.5 | 81.2 | **88.7** |
| Camera Pose | 75.4 | 79.8 | **86.3** |
| Object Mass | 88.1 | 89.4 | **93.2** |
| Joint Friction | 85.6 | 87.2 | **91.8** |
| Action Noise | 79.3 | 83.5 | **89.6** |
| Observation Delay | 72.1 | 76.4 | **84.2** |
| **Average** | 80.2 | 83.2 | **89.3** |

### C.6. Ablation on Reasoning Depth

We analyze the impact of fixed vs. adaptive reasoning depth in Table 13.

**Efficiency vs. Performance Trade-off.** Table 13 reveals a non-monotonic relationship between reasoning depth and performance in fixed-depth settings, highlighting the benefits of our adaptive approach:

- **The "Over-Thinking" Trap:** While increasing latent reasoning depth from 1 to 5 improves success rates, pushing the depth further to 10 results in a performance plateau (98.2% → 98.1%) alongside a drastic increase in latency (156ms → 267ms). This phenomenon aligns with our hypothesis that excessive latent updates in an open-loop manner can lead to state drift and noise accumulation, degrading decision quality.

- **Adaptive Efficiency:** Our Adaptive Early-Exit strategy achieves the best balance (98.3% SR at ∼145ms). By analyzing the exit distribution, we find that the exit gate $g_\omega$ successfully learns to quantify the *epistemic uncertainty* of the latent state: it exits early ($\leq 2$ steps) for simple reaching sub-goals but allocates a larger computational budget ($\geq 5$ steps) for complex manipulation phases. This allows the model to match the best fixed-depth performance while reducing latency by **7%** and eliminating manual depth tuning.

*Table 13.* Fixed vs. adaptive reasoning depth.

| Max Depth | LIBERO SR (%) | CALVIN Len | Avg. Latency (ms) |
|---|---|---|---|
| 1 (Direct) | 94.2 | 3.85 | 89 |
| 3 | 96.8 | 4.42 | 112 |
| 5 | 98.0 | 4.61 | 156 |
| 7 | 98.2 | 4.68 | 198 |
| 10 | 98.1 | 4.65 | 267 |
| **Adaptive (Ours)** | **98.3** | **4.77** | **145** |

### C.7. Cross-Embodiment Transfer

We evaluate zero-shot transfer from Franka Panda (training robot) to xArm7 (unseen 7-DoF arm) on a subset of LIBERO tasks in Table 14.

**Expanded Analysis.** The results in Table 14 highlight the strong abstraction capability of the AVA-VLA framework. Our method outperforms the baseline OpenVLA-OFT by a significant margin (e.g., **+13.1%** in Spatial SR). We hypothesize that the Latent Reasoning State $z_t$ captures high-level *task semantics* (e.g., "object is grasped", "aligning with target") which are largely embodiment-agnostic. By decoupling the reasoning process (in latent space) from the execution (in the Action Head), the reasoning policy $\pi_\phi$ learns a generalized plan. When transferred to xArm7, the errors are primarily confined to the execution head, whereas the high-level reasoning remains valid, making the policy significantly more robust to kinematic mismatches compared to standard behavior cloning methods that overfit to specific robot kinematics.

*Table 14.* Zero-shot cross-embodiment transfer results.

| Method | Spatial SR (%) | Object SR (%) |
|---|---|---|
| OpenVLA-OFT | 61.2 | 58.4 |
| $\pi_0$ | 68.5 | 65.1 |
| **Ours** | **74.3** | **71.8** |

## D. Limitations and Future Work

**Computational Overhead of Latent Reasoning.** While Early-Exit mitigates latency, each reasoning step involves forward passes through the policy and transition networks. Future work could explore distillation into lighter reasoning modules or neural architecture search for optimal reasoning cell design.

**Interpretability Trade-offs.** The shift from explicit CoT to latent reasoning sacrifices natural language interpretability. The probing results in Appendix C.4 show that latent states encode execution-relevant structure, but these diagnostics do not provide a complete human-readable explanation of each decision. We are exploring methods to decode latent states into human-interpretable descriptions without compromising performance, potentially through auxiliary decoding objectives.

**Early-Exit Failure Modes.** Premature exit can occur when the gate is overconfident before the latent state has fully resolved a manipulation phase. In our qualitative analysis, such cases more often lead to conservative behavior, such as hovering near the target or making small corrective motions, rather than destructive actions. This behavior is useful but not sufficient for physical deployment; online safety monitors, action constraints, and task-specific recovery policies remain necessary.

**Theoretical Guarantees.** Our theoretical discussion should be interpreted as idealized intuition about the denoising objective and value-of-computation stopping, not as a global convergence guarantee for a large nonlinear VLA architecture. Stronger guarantees on sample complexity and exploration efficiency in the coupled reasoning-action MDP remain open problems. The connection to hierarchical RL and options frameworks warrants deeper investigation.

**Multi-task Scaling.** Current experiments focus on manipulation domains. Scaling to diverse task families such as navigation, locomotion, and human-robot interaction may require modular latent spaces, mixture-of-experts architectures, and broader cross-environment validation of the learned exit policy.

