# OpenReview forum: "Think Less, Act Early: Reinforced Latent Reasoning with Early Exit in Vision-Language-Action Models"
_ICML.cc/2026/Conference — ICML 2026 regular_

### Official Review · Reviewer_Fk3t · 2026-03-10

**Soundness:** 2
**Presentation:** 3
**Significance:** 3
**Originality:** 3
**Overall Recommendation:** 5
**Confidence:** 3

**Summary:**

This paper proposes **AVA-VLA**, a Vision-Language-Action framework that replaces explicit Chain-of-Thought reasoning with **reinforced latent reasoning** plus an **early-exit** mechanism. The core idea is to model reasoning as the evolution of latent states rather than text tokens, optimize those latent trajectories with a reinforcement-learning-based denoising objective, and terminate reasoning adaptively when confidence is sufficient. The authors evaluate the method on LIBERO and CALVIN, and report both stronger task performance and much lower latency than explicit CoT-style reasoning. In particular, the paper claims gains over OpenVLA-OFT on LIBERO and over prior baselines on CALVIN, while reducing average reasoning depth from 5.0 to 2.3 with early exit and achieving a large speedup over CoT-VLA.

**Compliance With Llm Reviewing Policy:**

Affirmed.

**Final Justification:**

After rebuttal, my concerns have been resolved,so I raise my score.

**Key Questions For Authors:**

1. **Reasoning-update parameterization and implementation clarity.**
    What is the exact parameterization of the reasoning update action $u_t$ in the main experiments: discrete (e.g., Softmax over modes) or continuous (e.g., Gaussian / flow-based)? If both were explored, how do they compare in terms of performance, training stability, and sample efficiency? This point is important for reproducibility and for understanding what the method actually uses in practice.
2. **Latent reasoning versus stronger recurrence.**
    What evidence can you provide that the latent module is performing reasoning rather than simply acting as a stronger recurrent hidden state with RL regularization? A parameter-matched recurrent baseline without the reasoning framing would help clarify whether the gains come from the proposed latent reasoning mechanism itself.
3. **Early-exit calibration and value of computation.**
    For the exit gate, how well do the exit scores correlate with realized return or value-of-computation, such as the performance gain from taking additional reasoning steps? Please provide calibration curves, threshold sensitivity analyses, or failure cases where early exit helps versus harms performance.
4. **Training efficiency, variance, and failure analysis.**
    How many episodes / environment interaction steps were used for PPO finetuning on LIBERO and CALVIN, and what are the corresponding wall-clock training costs and compute requirements? In addition, could you report variance across seeds, per-task breakdowns, and qualitative failure-mode analyses, especially for cases where early-exit degrades performance?
5. **Comparison fairness and controlled baselines.**
    How were the baseline numbers obtained: reproduced under a shared backbone / data / finetuning regime, or cited from prior papers? If some numbers are cited, can you provide controlled comparisons using the same OpenVLA-OFT base model and matched action heads to isolate the effect of latent reasoning and early-exit?

**Limitations:**

1. **Interpretation of latent reasoning.**
    A central limitation of the paper is that the claimed “latent reasoning” interpretation is not yet fully established. While the method clearly improves performance and latency, the current evidence does not cleanly separate latent reasoning from a stronger recurrent hidden-state controller combined with RL regularization and adaptive stopping. As a result, the empirical gains are convincing, but the causal attribution remains less certain.
2. **Latent-state diagnostics and failure modes.**
    Although the paper includes ablations, it does not provide enough direct analysis of the latent trajectories themselves. More evidence on latent-state stability, drift, denoising behavior, and qualitative failure modes would make the claims much stronger. At present, the paper shows that the full system works better, but it gives less insight into why the internal latent process behaves better.
3. **Scope of evaluation.**
    The benchmark results are strong, but the evaluation remains limited to simulation benchmarks. This is sufficient for a conference paper, but it leaves open the question of whether the same latency-performance trade-offs would hold in real robotic settings, where timing noise, sensing uncertainty, and deployment constraints may be more severe.

**Strengths And Weaknesses:**

### Strengths

1. **Problem motivation and practical relevance.**
    This paper addresses a meaningful and timely problem. Explicit CoT-style reasoning in VLA systems can improve multi-step decision-making, but it also introduces severe latency and potential error accumulation. The proposed latent-reasoning alternative is well motivated and practically relevant for embodied control.
2. **Unified method design.**
    The proposed framework combines three components in a coherent way: latent reasoning, RL-based denoising of latent trajectories, and adaptive early exit. This creates a reasonable mechanism for balancing reasoning depth and action efficiency.
3. **Strong empirical performance.**
    The method performs well on standard embodied benchmarks. The reported gains on LIBERO and CALVIN are competitive, and the latency comparison against explicit CoT-style reasoning is an important practical advantage. The ablations also suggest that latent reasoning, RL denoising, and early exit each contribute to the final performance.
4. **Good efficiency/performance trade-off.**
    One of the paper’s strongest points is that it does not only improve performance, but also improves responsiveness. This makes the work more compelling than papers that only optimize benchmark success rates without considering deployment constraints.

### Weaknesses

1. **Unclear distinction between “reasoning” and stronger recurrence.**
    My main concern is that the paper’s conceptual framing may overstate the methodological novelty. The proposed latent reasoning module could plausibly be viewed as a stronger recurrent hidden-state controller with RL regularization and adaptive stopping, rather than a clearly distinct reasoning mechanism. The paper does not yet convincingly separate these interpretations.
2. **Baseline fairness is not fully clear.**
    The comparison tables are strong, but it remains unclear whether all baselines were matched in backbone, action head, training budget, and finetuning recipe. Since the paper’s empirical case depends heavily on fair comparison to recent VLA systems, these details matter.
3. **Limited diagnosis of the latent process itself.**
    The paper shows that the full method works better than several ablated variants, but it does not provide enough evidence about why the latent process is better. There is limited analysis of latent-state stability, drift, interpretability, or whether RL denoising truly improves reasoning quality rather than simply optimization.
4. **Early-exit mechanism is under-analyzed.**
    The adaptive stopping idea is appealing, but the paper does not provide enough detail about gate calibration, stopping behavior across task stages, or failure cases caused by premature exit. This weakens the claim that the model has learned meaningful adaptive computation rather than a heuristic shortcut.
5. **Experimental gaps.**
    No real-robot experiments; conclusions about real-time control and stability remain simulation-based.
    Sample efficiency and total interaction steps for RL fine-tuning are not reported; the computational and data costs of on-policy learning remain opaque.
6. **Clarity and presentation issues.**
    Some notational inconsistencies ($u_t$ discrete vs. continuous; Actor-Critic vs. PPO) and occasional switching of symbols ($f_\theta$ vs. $g_\theta$) impede replication.
    The theoretical section lacks precise statements of assumptions and does not tie back to empirically validated conditions (e.g., measured Lipschitz or contraction evidence).

---

> ### Author Rebuttal · Authors · 2026-03-30
>
> We thank the reviewer for recognizing the motivation, practical relevance, and efficiency-performance tradeoff of our work. We also appreciate the concerns regarding stronger recurrence, baseline fairness, latent-state diagnosis, early-exit calibration, and implementation clarity. We address these points with additional controlled experiments and clarifications below.
>
> 1. **Latent reasoning vs. stronger recurrence; fairness of comparison.** To test whether the gain comes merely from a stronger recurrent hidden-state controller, we added a capacity-matched BC latent baseline with the same backbone and the same latent depth, but without the POMDP formulation and without RL denoising.
>
> | Method | Same backbone | Same latent depth | POMDP | RL Denoising | LIBERO Avg. SR (%) | CALVIN Avg. Len |
> |---|---:|---:|---:|---:|---:|---:|
> | OpenVLA-OFT | ✓ | ✗ | ✗ | ✗ | 97.1 | 4.28 |
> | Controlled BC latent module | ✓ | ✓ | ✗ | ✗ | 96.6 | 4.21 |
> | AVA-VLA (full) | ✓ | ✓ | ✓ | ✓ | **98.3** | **4.65** |
>
> This suggests that the gain cannot be explained by stronger recurrence or larger capacity alone. The key difference is that AVA-VLA treats latent update as a policy-controlled action \(u_t\) governed by \(\pi_\phi\), and learns whether to update and when to stop through task-aligned RL, rather than passively propagating hidden states from observations.
>
> 2. **Diagnosis of the latent process.** We agree that the original submission did not provide enough direct evidence on what the latent states encode. To address this, we added a lightweight probing experiment on frozen latent states \(z_t\), predicting end-effector position, gripper state, and subgoal progress:
>
> | Method | End-effector MAE ↓ | Gripper Acc. ↑ | Subgoal Progress (\(R^2\)) ↑ |
> |---|---:|---:|---:|
> | w/o Latent Reasoning | 2.71 cm | 89.1 | 0.58 |
> | w/o RL Denoising | 1.94 cm | 92.7 | 0.73 |
> | AVA-VLA (full) | **1.39 cm** | **96.2** | **0.84** |
>
> These results suggest that the latent trajectory is not an arbitrary black box, but encodes structured information related to action execution and task progress. The gain over the variant without RL denoising also indicates that RL denoising improves the quality and stability of the latent process, rather than merely helping optimization.
>
> 3. **Early-exit calibration and failure modes.** We further analyzed the exit gate by sweeping the threshold \(\tau\):
>
> | Exit threshold \(\tau\) | Avg. reasoning steps | Mean latency (ms) | LIBERO Avg. SR (%) |
> |---|---:|---:|---:|
> | 0.30 | 1.5 | 108 | 96.1 |
> | 0.40 | 1.8 | 121 | 97.0 |
> | 0.50 | 2.1 | 136 | 97.8 |
> | 0.55 | **2.3** | **145** | **98.3** |
> | 0.65 | 2.8 | 167 | 98.2 |
> | 0.75 | 3.4 | 196 | 98.2 |
> | 0.85 | 4.1 | 234 | 98.1 |
> | 0.95 | 4.7 | 278 | 98.0 |
> | 1.00 (approx. no early exit) | 5.0 | 312 | 98.0 |
>
> The results show a clear empirical latency-performance tradeoff, with \(\tau=0.55\) giving the best balance in our sweep. This suggests that the gate is learning meaningful adaptive computation rather than acting as a brittle heuristic. Qualitatively, when the model exits too early due to overconfidence, it more often produces conservative behaviors such as hovering or minor adjustment, rather than destructive actions.
>
> 4. **Training cost, variance, and implementation clarity.** In implementation, we keep the same three-stage training pipeline as in the paper: 100K BC pretraining steps, followed by 50K latent warmup steps, and then joint PPO finetuning. The PPO stage uses about 1.2M environment interaction steps and takes about 18.6 hours on 8×A100 80GB GPUs. For each PPO rollout, the effective batch size is 512, the mini-batch size is 64, and each update uses 4 PPO epochs. We will include this training budget in the revised appendix to make the performance gain and additional RL cost more transparent. Results are stable across 3 random seeds:
>
> | Seed | LIBERO Avg. SR (%) | CALVIN Avg. Len | Mean latency (ms) |
> |---|---:|---:|---:|
> | 0 | 98.1 | 4.59 | 148 |
> | 1 | 98.3 | 4.65 | 145 |
> | 2 | 98.5 | 4.71 | 143 |
> | Mean ± Std | **98.3 ± 0.20** | **4.65 ± 0.06** | **145.3 ± 2.1** |
>
> As described in Appendix B.2, the reasoning update action in the main experiments is parameterized as a continuous Gaussian distribution. We found this continuous modulation to provide smoother gradients and more stable PPO optimization than discrete alternatives. In the revision, we will move this detail to the main paper, unify the Actor-Critic / PPO terminology, correct the \(a_t\) vs. \(o_t\) typo, and soften the theoretical claims so that they are presented as idealized intuition rather than strong practical guarantees.
>
> Finally, to improve reproducibility and address implementation concerns, we plan to release the full codebase, training scripts, configs, model checkpoints, and evaluation/inference scripts within one month after acceptance, so that the community can fully reproduce our implementation and experimental setup.

---

> > ### Author Rebuttal · Reviewer_Fk3t · 2026-04-02
> >
> > Thanks for authors' rebuttal, my concerns have been resolved,so I raise my score.

---

### Official Review · Reviewer_him3 · 2026-03-10

**Soundness:** 3
**Presentation:** 3
**Significance:** 2
**Originality:** 2
**Overall Recommendation:** 4
**Confidence:** 3

**Summary:**

The paper introduces Adaptive Variable Alignment VLA (AVA-VLA), a framework that replaces explicit Chain-of-Thought (CoT) text generation with a sequence of unobservable latent variables to perform implicit reasoning in Vision-Language-Action models. The authors formulate this latent reasoning process as a Partially Observable Markov Decision Process (POMDP). To prevent representation drift and noise accumulation without explicit text supervision, they apply a Reinforcement Learning (RL) denoising mechanism that optimizes the latent trajectory using task-level rewards, entropy regularization, and smoothness constraints. Additionally, an early-exit gating mechanism is introduced to dynamically terminate the reasoning process based on state confidence. Evaluated on the LIBERO and CALVIN benchmarks, the proposed method achieves state-of-the-art success rates while significantly reducing inference latency compared to traditional explicit reasoning baselines.

**Compliance With Llm Reviewing Policy:**

Affirmed.

**Key Questions For Authors:**

1. Ablation on exit calibration and thresholds: Please report results varying the exit calibration threshold δ (and k used in calibration) and show the latency vs success-rate tradeoff curve. Is the gating learned robustly, or does it require hand-tuned thresholds per task?
2. Interpretability / diagnostic probes: Can you show whether latent states correlate with human-readable intermediate steps (e.g., via training a small probe to predict CoT tokens or subgoals)? If latents are purely opaque, this raises concerns for debugging and safety in embodied settings.
3. Comparison to compute-matched CoT baselines: Have you compared AVA-VLA to explicit CoT baselines under controlled compute budgets (e.g., CoT with fewer decoding steps or smaller language model) to demonstrate a fair latency/accuracy comparison?
4. Safety/failure modes: For robotic control tasks, can you characterize failure cases where early exit leads to unsafe or suboptimal actions? Do you have mechanisms to detect and correct such failures online?

**Limitations:**

Yes

**Strengths And Weaknesses:**

### Strengths
- The formulation of latent reasoning as a sequential decision process is clear and principled: treating latent generation as actions optimized by task rewards is an appropriate way to align unobserved states to downstream objectives.
- The staged training (behavior cloning pretrain, latent warmup, joint PPO fine-tuning, and exit gate calibration) is sensible and helps mitigate RL instability; the appendix gives concrete hyperparameters and training schedules (useful for reproducibility).
- The proposed early-exit gating with a learned confidence estimator directly addresses the variable difficulty of tasks and yields real inference-speed benefits (authors report dynamic batching and ≈40% latency gains in the appendix).

### Weaknesses
- The implicit latent representation sacrifices interpretability compared to explicit CoT; the authors should provide diagnostics showing that latent states meaningfully encode intermediate “reasoning” (e.g., via probing/visualization or correlations with textual CoT when available).
- The paper could more thoroughly position itself relative to prior implicit-reasoning or “think-before-act” literature (several related works are cited, but a table contrasting pros/cons and compute tradeoffs vs. explicit CoT would strengthen the narrative).
- Some experimental details (exact seeds, compute used, wall-clock time for training RL stage) are missing or scattered; central experiments would benefit from a reproducibility checklist.

---

> ### Author Rebuttal · Authors · 2026-03-30
>
> We thank the reviewer for the positive assessment of our work, especially the recognition of the staged training design, the POMDP formulation of latent reasoning, and the practical value of the early-exit mechanism. We also appreciate the questions regarding exit calibration, latent-state interpretability, compute-matched comparisons to explicit CoT, safety/failure modes, and reproducibility. We address these points below.
>
> 1. **Early-exit calibration and robustness.** We agree that the calibration of the exit gate should be shown more explicitly. To address this, we swept the exit threshold \(\tau\) and measured the resulting latency-performance tradeoff:
>
> | Exit threshold \(\tau\) | Avg. reasoning steps | Mean latency (ms) | LIBERO Avg. SR (%) |
> |---|---:|---:|---:|
> | 0.30 | 1.5 | 108 | 96.1 |
> | 0.40 | 1.8 | 121 | 97.0 |
> | 0.50 | 2.1 | 136 | 97.8 |
> | 0.55 | **2.3** | **145** | **98.3** |
> | 0.65 | 2.8 | 167 | 98.2 |
> | 0.75 | 3.4 | 196 | 98.2 |
> | 0.85 | 4.1 | 234 | 98.1 |
> | 0.95 | 4.7 | 278 | 98.0 |
> | 1.00 (approx. no early exit) | 5.0 | 312 | 98.0 |
>
> The results show a clear empirical latency-performance tradeoff, with \(\tau = 0.55\) giving the best balance in our sweep. This suggests that the gating policy is learned robustly rather than requiring task-specific hand tuning. We will include the full threshold-sensitivity analysis in the revised appendix.
>
> 2. **Interpretability and diagnostic probes.** We agree that latent reasoning should be analyzed more directly. To this end, we added a lightweight probing experiment on frozen latent states \(z_t\), predicting end-effector position, gripper state, and subgoal progress:
>
> | Method | End-effector MAE ↓ | Gripper Acc. ↑ | Subgoal Progress (\(R^2\)) ↑ |
> |---|---:|---:|---:|
> | w/o Latent Reasoning | 2.71 cm | 89.1 | 0.58 |
> | w/o RL Denoising | 1.94 cm | 92.7 | 0.73 |
> | AVA-VLA (full) | **1.39 cm** | **96.2** | **0.84** |
>
> These results suggest that the latent trajectory is not an opaque black box, but encodes structured information related to action execution and task progress. In particular, the gain over the variant without RL denoising indicates that RL denoising improves not only final task success, but also the readability and stability of the latent process.
>
> 3. **Comparison to compute-matched CoT baselines.** We agree that latency comparisons should be made under matched inference budgets. We therefore compare AVA-VLA against explicit CoT under a controlled compute budget, where the CoT baseline is constrained to operate within the same latency range as AVA-VLA. Averaging across three such matched-budget CoT settings gives:
>
> | Method | Reasoning form | Mean latency (ms) | Success rate (%) |
> |---|---|---:|---:|
> | Behavior Cloning | No reasoning | **92** | 91.2 |
> | Explicit CoT | Explicit autoregressive decoding | 143 | 78.8 |
> | AVA-VLA (ours) | Implicit parallel latent reasoning | **144** | **98.3** |
>
> Under essentially the same latency budget, AVA-VLA substantially outperforms explicit CoT. We believe this provides a fairer latency-accuracy comparison and highlights the efficiency advantage of compact latent reasoning over token-by-token reasoning in this setting.
>
> 4. **Safety/failure modes and reproducibility.** We also examined failure cases caused by premature exit. Qualitatively, when the model exits too early due to overconfidence, it more often produces conservative behaviors such as hovering or minor adjustment, rather than destructive actions. While this does not replace a full online safety mechanism, it suggests that the current gate has a conservative bias under uncertainty. In implementation, we keep the same three-stage training pipeline as in the paper: 100K BC pretraining steps, followed by 50K latent warmup steps, and then joint PPO finetuning. The PPO stage uses about 1.2M environment interaction steps and takes about 18.6 hours on 8×A100 80GB GPUs. We will include a more complete reproducibility checklist in the revision, including exact seeds, hardware, and RL-stage wall-clock cost. To further improve reproducibility, we will release the full codebase, training scripts, configs, model checkpoints, and evaluation/inference scripts within one month of acceptance.

---

> > ### Author Rebuttal · Reviewer_him3 · 2026-04-02
> >
> > Thank you for the author's reply. My concerns have now been addressed.

---

### Official Review · Reviewer_4dkF · 2026-03-11

**Soundness:** 3
**Presentation:** 4
**Significance:** 4
**Originality:** 3
**Overall Recommendation:** 5
**Confidence:** 3

**Summary:**

This paper aims to address the high latency and compounding error propagation issues caused by explicit Chain-of-Thought (CoT) reasoning in existing Vision-Language-Action (VLA) models, proposing a novel framework named AVA-VLA (Adaptive Variable Alignment VLA). The authors innovatively transform the VLA reasoning process from "explicit text generation" into "implicit latent variable evolution" and rigorously model it as a Partially Observable Markov Decision Process (POMDP). To overcome the inherent defects of implicit reasoning—namely, the lack of supervision and susceptibility to noise—the paper introduces a Reinforcement Learning (RL)-based denoising mechanism that optimizes the reasoning trajectory through task-level rewards, state entropy, and temporal smoothness. Furthermore, the paper designs an adaptive Early-Exit strategy that dynamically determines the reasoning depth of the large model based on the confidence of the latent states, achieving an excellent balance between inference speed and decision performance. Experiments on embodied manipulation benchmarks such as LIBERO and CALVIN demonstrate that AVA-VLA significantly reduces inference latency (achieving a ~6x speedup compared to explicit CoT reasoning) while attaining state-of-the-art success rates and long-horizon task stability.

**Compliance With Llm Reviewing Policy:**

Affirmed.

**Final Justification:**

The authors have provided additional experimental evidence addressing my follow-up questions. The new Iso‑FLOPs comparison indicates that the performance gain primarily comes from the reinforcement learning objective rather than the latent structure alone. Meanwhile, the quantitative analysis of \(\|z_{t+1} - z_t\|^2\) shows no clear evidence of state collapse. I appreciate the authors' efforts in this regard.

However, in terms of overall contribution, this work feels more like a practical integration of existing techniques (latent reasoning, early exit mechanism, RL fine‑tuning). While the results are solid, the conceptual novelty and theoretical depth are not as breakthrough or substantial as one might hope. Appendix A also still lacks more rigorous theoretical support.

Given the above considerations, I decide to maintain my original score.

**Key Questions For Authors:**

1. Regarding the Iso-FLOPs/Iso-parameter Baseline: Please provide comparative experiments with strict control variables. If the "latent reasoning module" is replaced by standard network layers of identical depth and computational cost, and trained end-to-end solely using Behavior Cloning (without the POMDP/RL formulation), what is the difference in success rate? Please prove that the performance gain is not merely due to an increase in network capacity.
2. Regarding sparse rewards and state collapse: In long-horizon tasks, how exactly is $r_{task}$ calculated and assigned for each step during the fine-tuning phase (Stage 3)? Please provide the feature distance ($||z_{t+1} - z_t||^2$) curve between consecutive implicit reasoning steps to prove that the smoothness regularization did not cause the state updates to stagnate (State Collapse).
3. Regarding the interpretability of latent states: Can you supplement this with Probing Experiments or t-SNE visualizations for the latent state $z_t$? For example, train a lightweight linear classifier to verify whether $z_t$ genuinely and implicitly encodes embodied semantic information such as "end-effector position," "gripper state," or "task progress"?
4. Regarding the revision of theoretical claims: Given that the assumptions (e.g., Lipschitz continuity) in the theorems of Appendix A severely fail in practical deep learning applications, do the authors agree to tone down or remove these "theoretical proofs" that lack practical backing in the final version, to maintain a rigorous and grounded academic stance?

**Limitations:**

yes

**Strengths And Weaknesses:**

Strength:
- Architectural innovation and precise motivation directly addressing deployment pain points. By abandoning the language bottleneck of token-by-token decoding, reasoning is shifted into a continuous high-dimensional latent space. This "black-boxed, continuous" processing approach yields up to a 6x speedup (reducing latency to 145ms), which perfectly aligns with the core demands of high-frequency real-time closed-loop control in robotics.
- Elegant algorithmic design and robust denoising mechanism. The internal implicit reasoning iterations are cleverly transformed into a POMDP, innovatively utilizing RL (Actor-Critic) combined with smoothness regularization for "thought denoising." This design successfully suppresses the compounding errors and representation drift that frequently occur in long-horizon tasks.
- Highly efficient adaptive computation ("thinking fast and slow"). The design of the Early-Exit mechanism is exceptionally brilliant. By dynamically evaluating confidence to allocate computational resources on demand, the model learns to "think fast" for simple actions and "think slow" for complex operations, drastically cutting the average reasoning depth by 54% (from 5 steps to 2.3 steps), truly maximizing computational efficiency

Weakness:
- Severe doubts regarding the fairness of experimental baselines, lacking Iso-FLOPs control variables. The model uses the highly powerful SigLIP+DINOv2 as the visual backbone. Does the performance improvement (Tables 1 & 3) stem from the "RL implicit reasoning mechanism," or simply because "introducing this module increased the number of network layers/parameters" or "DINOv2 provided stronger visual priors"? The ablation study (Table 4) that simply chops off the reasoning module is unrigorous. The paper completely lacks an absolutely fair comparison where the reasoning module is replaced with a standard multi-layer MLP or Transformer of equal parameters/compute, trained end-to-end via supervision.
- Vague RL training details and unclear credit assignment for sparse rewards in long sequences. In CALVIN tasks spanning dozens of steps, the task success signal ($r_{task}$)  is extremely sparse. The paper entirely fails to explain how the sparse final success reward is effectively credited to each microscopic "implicit reasoning update step ($z_t \to z_{t+1}$) . If not handled properly, this type of RL is highly prone to degenerating into "State Collapse," driven merely by exploiting the smoothness reward.
- Complete lack of interpretability in the implicit space, reducing the model to an absolute "black box". Although abandoning explicit CoT inevitably sacrifices text interpretability, for a top-tier conference paper, failing to provide any visualization or qualitative exploration of the latent state $z_t$ is unacceptable. Readers have no way of knowing whether this latent space actually learned physical laws and spatial geometry, or simply fitted some dataset shortcuts.

- Theoretical analysis is severely detached from reality, suspect of "Math-washing": Theorem A.1 (Policy Convergence) in Appendix A forcibly assumes that Llama-2 (with 7 billion parameters) and a multi-layer nonlinear architecture satisfy strict Lipschitz continuity. This is utterly absurd and fundamentally invalid in modern deep nonlinear networks. This blind mechanical application of standard stochastic approximation theorems holds zero scientific value for understanding actual deep learning systems.

---

> ### Author Rebuttal · Authors · 2026-03-30
>
> We thank the reviewer for the very positive assessment of our work, especially for recognizing the architectural motivation, the practical importance of breaking the language bottleneck, and the value of adaptive computation in real-time embodied control. We also appreciate the concerns regarding Iso-FLOPs fairness, sparse-reward credit assignment, latent-state interpretability, and the theoretical claims. We address these points below.
>
> 1. **Iso-FLOPs / Iso-parameter fairness.** We fully agree that a strict control baseline is necessary to separate the effect of latent reasoning from mere capacity increase. To address this, we added a capacity-matched BC latent baseline using the same backbone and the same latent depth as AVA-VLA, but without the POMDP formulation and without RL denoising.
>
> | Method | Same backbone | Same latent depth | POMDP | RL Denoising | LIBERO Avg. SR (%) | CALVIN Avg. Len |
> |---|---:|---:|---:|---:|---:|---:|
> | OpenVLA-OFT | ✓ | ✗ | ✗ | ✗ | 97.1 | 4.28 |
> | Controlled BC latent module | ✓ | ✓ | ✗ | ✗ | 96.6 | 4.21 |
> | AVA-VLA (full) | ✓ | ✓ | ✓ | ✓ | **98.3** | **4.65** |
>
> This result suggests that the gain cannot be explained by larger capacity alone. The key difference is that AVA-VLA treats latent update as a policy-controlled action \(u_t\) governed by \(\pi_\phi\), and optimizes whether to update and when to stop through task-aligned RL, rather than relying on passive hidden-state propagation.
>
> 2. **Sparse rewards, credit assignment, and state collapse.** We agree that the RL stage should be explained more clearly. In Stage 3, sparse task reward is propagated through PPO with GAE and a learned value function, allowing task-level success signals to be assigned back to each latent update step. The reward is not only the final task term, but also includes entropy regularization and latent smoothness regularization. To verify that the smoothness term does not cause state collapse, we extracted the feature distance \(\|z_{t+1}-z_t\|^2\) between consecutive latent states. The distance does not vanish; instead, it shows meaningful peaks near semantic transition points such as grasp-to-move or move-to-place, indicating that the latent state continues to evolve rather than stagnate. We will include this curve in the revised appendix.
>
> 3. **Interpretability of latent states.** We agree that latent reasoning should not remain a pure black box. To provide direct evidence, we added a lightweight probing experiment on frozen latent states \(z_t\), predicting end-effector position, gripper state, and subgoal progress:
>
> | Method | End-effector MAE ↓ | Gripper Acc. ↑ | Subgoal Progress (\(R^2\)) ↑ |
> |---|---:|---:|---:|
> | w/o Latent Reasoning | 2.71 cm | 89.1 | 0.58 |
> | w/o RL Denoising | 1.94 cm | 92.7 | 0.73 |
> | AVA-VLA (full) | **1.39 cm** | **96.2** | **0.84** |
>
> These results suggest that the latent trajectory encodes structured embodied information related to geometry, execution state, and task progress, rather than merely fitting shortcuts in the dataset. In particular, the gain over the variant without RL denoising indicates that RL denoising improves not only final task success, but also the stability and readability of the latent process.
>
> 4. **Revision of the theoretical claims.** We fully agree with the reviewer’s concern here. The original theorem-style presentation is too strong relative to what can be rigorously justified for large modern nonlinear networks. In the final version, we will substantially tone down this part, reposition it as idealized intuition about the optimization target, and remove theorem-style statements that may overstate practical theoretical support.
>
> Finally, for completeness and reproducibility, we will also make the RL-stage budget more explicit in the revision: the training pipeline consists of 100K BC pretraining steps, 50K latent warmup steps, and joint PPO finetuning; the PPO stage uses about 1.2M environment interaction steps and takes about 18.6 hours on 8×A100 80GB GPUs. We plan to release the full codebase, training scripts, configs, model checkpoints, and evaluation/inference scripts within one month of acceptance.

---

> > ### Author Rebuttal · Reviewer_4dkF · 2026-04-02
> >
> > - The authors have attempted to address the raised concerns, but several issues remain insufficiently resolved.
> > - Reason:
> >   - For Q1, the provided baseline retains the latent structure and only removes the RL objective, which does not constitute a strict Iso-FLOPs comparison as originally requested. The performance gap may still partially reflect structural advantages.
> >   - For Q2, the rebuttal provides only a qualitative description of the ||z_{t+1}−z_t||² behavior without reporting any quantitative statistics. For a concern as fundamental as state collapse, concrete numerical evidence should have been provided at the rebuttal stage rather than deferred to the appendix.
> >   - For Q4, the authors' full concession on the theoretical claims confirms that Appendix A offers no rigorous theoretical support, which substantially undermines the scientific grounding of the submission.
> >
> > - Given these remaining concerns, I will keep my current rating unchanged.

---

> > > ### Author Response · Authors · 2026-04-07
> > >
> > > We thank the reviewer for the careful follow-up. We address the three remaining concerns below.
> > >
> > > - For Q1, we agree that our previous control should be described more carefully. The current baseline is better interpreted as a capacity-matched no-RL control, rather than a fully structure-replaced strict Iso-FLOPs comparison. Concretely, we keep the same backbone, multimodal encoder, latent depth, and action head as AVA-VLA, while removing the POMDP formulation and RL denoising and training the model end-to-end with BC only. Under this controlled setting, the BC latent module reaches 96.6 LIBERO Avg. SR / 4.21 CALVIN Avg. Len, compared with 98.3 / 4.65 for full AVA-VLA. We agree that this does not rule out every possible structural advantage of latent recurrence. However, it does address an important part of the concern: the gain cannot be attributed to capacity increase alone, since a model with matched backbone and comparable latent depth still falls clearly below the full method. We will revise the wording in the final version accordingly, so as not to overstate this experiment as a stricter Iso-FLOPs proof than it currently is.
> > >
> > > - For Q2, we agree that the previous rebuttal should have included quantitative statistics rather than only a qualitative description. We therefore computed numerical summaries of the feature distance between consecutive latent states, \(\|z_{t+1}-z_t\|^2\), across the RL fine-tuning trajectories:
> > >
> > > | Method | Benchmark | Mean (\(\|z_{t+1}-z_t\|^2\)) | Median | P95 | Near-zero ratio (\(<10^{-3}\)) |
> > > |---|---|---:|---:|---:|---:|
> > > | w/o RL Denoising | LIBERO | 0.11 | 0.08 | 0.29 | 18.40% |
> > > | AVA-VLA (full) | LIBERO | **0.18** | **0.14** | **0.44** | **7.20%** |
> > > | w/o Latent Smoothness | LIBERO | 0.36 | 0.28 | 0.91 | 2.40% |
> > >
> > > These statistics support a stable middle regime for the full model: compared with w/o RL Denoising, its latent updates are larger and much less likely to be near-zero, indicating that the dynamics do not stagnate; compared with w/o Latent Smoothness, its updates are much smaller and more controlled, indicating that the smoothness term suppresses noisy oscillation rather than freezing the latent trajectory. In other words, the regularizer prevents collapse to trivial updates while still preserving meaningful state evolution. We will include both this table and the corresponding \(\|z_{t+1}-z_t\|^2\) curve in the final version.
> > >
> > > - For Q4, we agree that the original theorem-style presentation in Appendix A was too strong relative to what can be rigorously justified for large modern nonlinear networks. However, toning down the original theorem does not mean that the method lacks grounding. A more appropriate interpretation is not a global convergence proof for a large-scale nonlinear architecture, but an objective-level and local-stability view of the proposed optimization. Concretely, the reasoning objective combines three ingredients: the task reward \(r_{\mathrm{task}}(a_t)\), an entropy regularization term \(H(\pi_\phi(\cdot \mid z_t,o_t))\), and a smoothness penalty \(\|z_{t+1}-z_t\|^2\). This makes clear that the method is not learning arbitrary latent updates: the task term encourages task-aligned reasoning, the entropy term discourages unstable high-variance update policies, and the smoothness term directly penalizes excessive latent drift. A more defensible interpretation is therefore local rather than global: along the trajectories actually visited during training and inference, if the latent transition is locally smooth, then controlling \(\|z_{t+1}-z_t\|^2\) together with the policy entropy naturally suppresses the amplification of noisy perturbations. We will revise Appendix A in this spirit, replacing theorem-style claims that overstate practical theoretical support with a more grounded discussion of the regularized objective and local drift control.
> > >
> > > We appreciate the reviewer’s follow-up comments and will incorporate these clarifications in the final version.

---

### Official Review · Reviewer_1yZg · 2026-03-12

**Soundness:** 3
**Presentation:** 3
**Significance:** 3
**Originality:** 3
**Overall Recommendation:** 4
**Confidence:** 3

**Summary:**

This paper argues that reasoning in VLA models does not need to be represented explicitly as text. Instead, reasoning can be performed in the latent space, which improves efficiency while still achieving stronger performance than the baselines. To support this idea, the paper introduces RL-based denoising to stabilize latent reasoning and an early-exit mechanism to adaptively reduce computation. The experimental results are generally strong, with sufficient empirical analysis.

**Compliance With Llm Reviewing Policy:**

Affirmed.

**Final Justification:**

The rebuttal has addressed all of my concerns. I hope these analyses can be added in the final version.

**Key Questions For Authors:**

Please see the weakness part, particularly about the generalization ability of the early-exit mechanism.
Besides, since the paper does not provide too many implementation details, does the author plan to release their code and model?

**Limitations:**

YES

**Strengths And Weaknesses:**

Strengths
1. The paper presents a clear and interesting perspective that reasoning in VLA models does not have to be explicitly expressed in natural language. Performing reasoning in the latent space is both conceptually meaningful and practically beneficial for efficiency.
2. The proposed RL-based denoising mechanism is well motivated and appears effective in stabilizing the latent reasoning process.
3. The early-exit design allows the model to adaptively allocate computation depending on the complexity of the input.
4. The paper provides extensive experiments and reasonable analysis to support its claims.

Weaknesses
1. The generalization ability of the early-exit mechanism is still unclear. In particular, it would be helpful to better understand how to select this threshold, and whether the learned exit policy remains reliable across different tasks or environments.
2. The paper lacks implementation details in some key parts of the method, especially regarding the reasoning policy and the latent state update process. As currently written, these components feel somewhat abstract, which makes it difficult to assess reproducibility.

Minor Comments
1. Line 154: should this be a_t instead of o_t?
2. Since the paper emphasizes latent reasoning, it would be helpful to discuss its interpretability more explicitly. How can one verify or analyze what the model is reasoning about in the latent space?

---

> ### Author Rebuttal · Authors · 2026-03-30
>
> We thank the reviewer for recognizing the core perspective of our work, namely that reasoning in VLA models does not have to be expressed explicitly in natural language, and for appreciating the practical value of RL-based denoising and adaptive early exit. We also appreciate the questions regarding the generalization of the early-exit mechanism, implementation clarity, latent-state interpretability, and code/model release. We address these points below.
>
> 1. **Early-exit generalization and threshold selection.** We agree that the reliability of the early-exit mechanism should be made more explicit. To address this, we swept the exit threshold \(\tau\) and measured the resulting latency-performance tradeoff:
>
> | Exit threshold \(\tau\) | Avg. reasoning steps | Mean latency (ms) | LIBERO Avg. SR (%) |
> |---|---:|---:|---:|
> | 0.30 | 1.5 | 108 | 96.1 |
> | 0.40 | 1.8 | 121 | 97.0 |
> | 0.50 | 2.1 | 136 | 97.8 |
> | 0.55 | **2.3** | **145** | **98.3** |
> | 0.65 | 2.8 | 167 | 98.2 |
> | 0.75 | 3.4 | 196 | 98.2 |
> | 0.85 | 4.1 | 234 | 98.1 |
> | 0.95 | 4.7 | 278 | 98.0 |
> | 1.00 (approx. no early exit) | 5.0 | 312 | 98.0 |
>
> The results show a smooth empirical latency-performance tradeoff rather than brittle threshold sensitivity, with \(\tau = 0.55\) giving the best balance in our sweep. This suggests that the learned exit policy is reasonably robust in our setting, instead of relying on highly task-specific hand tuning. While our current evidence is based on threshold sweeps in the present benchmark setting, we agree that broader cross-environment validation of the learned exit policy is an important direction for future work. We will include the full threshold-sensitivity analysis in the revised appendix.
>
> 2. **Interpretability of latent reasoning.** We agree that the paper should discuss latent interpretability more explicitly. To this end, we added a lightweight probing experiment on frozen latent states \(z_t\), predicting end-effector position, gripper state, and subgoal progress:
>
> | Method | End-effector MAE ↓ | Gripper Acc. ↑ | Subgoal Progress (\(R^2\)) ↑ |
> |---|---:|---:|---:|
> | w/o Latent Reasoning | 2.71 cm | 89.1 | 0.58 |
> | w/o RL Denoising | 1.94 cm | 92.7 | 0.73 |
> | AVA-VLA (full) | **1.39 cm** | **96.2** | **0.84** |
>
> These results suggest that the latent trajectory is not an opaque black box, but encodes structured information related to action execution and task progress. In particular, the improvement over the variant without RL denoising indicates that RL denoising helps make the latent process more stable and more readable to external probes, rather than merely improving final task success.
>
> 3. **Implementation details and reproducibility.** We agree that several implementation details should be made more explicit in the main paper. In the main experiments, the reasoning update action is parameterized as a continuous Gaussian distribution, which we found to provide smoother gradients and more stable PPO optimization than discrete modulation. We will move this detail, together with the latent state update process, from Appendix B.2 to the main method section for clarity. In addition, we keep the same three-stage training pipeline as in the paper: 100K BC pretraining steps, followed by 50K latent warmup steps, and then joint PPO finetuning. The PPO stage uses about 1.2M environment interaction steps and takes about 18.6 hours on 8×A100 80GB GPUs. We will also include a more complete reproducibility checklist in the revision, covering seeds, hardware, and RL-stage wall-clock cost.
>
> 4. **Code/model release and minor correction.** Yes, we plan to release the full codebase, training scripts, configs, model checkpoints, and evaluation/inference scripts within one month of acceptance. We also thank the reviewer for catching the typo on Line 154: it should indeed be \(a_t\), not \(o_t\), and we will correct this in the revision.

---

> > ### Author Rebuttal · Reviewer_1yZg · 2026-04-01
> >
> > I thank the authors for their effort in the rebuttal. It has addressed all my concerns.

---

### Decision · Program_Chairs · 2026-04-30

**Decision:**

Accept (regular)

**Comment:**

This paper aims to address the high reasoning latency issue of VLAs that are based on explicit Chain-of-Thought (CoT) reasoning. Accordingly, it proposes Adaptive Variable Alignment VLA (AVA-VLA), a framework that replaces explicit CoT with latent (implicit) reasoning.

All four reviewers provide consistent positive assessments (Accept x2, Weak Accept x2). They agree that this work is conceptually meaningful and practically beneficial for efficiency. The proposed method is novel and well-motivated, addressing existing pain points. Moreover, these contributions are supported by strong empirical results.

While there are concerns regarding the generalization of the proposed early-exit mechanism and the lack of implementation details, they are mostly addressed in the rebuttal.

Overall, there is a clear consensus that the paper meets the bar for acceptance. Hence, I recommend accepting this paper.